# Research on Carbon Emission Quota of Railway in China from the Perspective of Equity and Efficiency

Yanan Guo [1] 🆔, Qiong Tong [1,*], Zhengjiao Li [2] and Yuhao Zhao [1]

1 School of Economics and Management, Beijing Jiaotong University, Beijing 100044, China
2 School of Electronic and Information Engineering, Beijing Jiaotong University, Beijing 100044, China
* Correspondence: qtong@bjtu.edu.cn; Tel.: +86-010-51683891

**Abstract:** Under the constraint of total carbon emissions, the allocation of carbon emission quotas of 18 railway bureaus in China is conducted to the realization of carbon emission reduction targets of China's railway transportation industry. This paper proposes a carbon emission quota model for China's railway industry from the perspective of equity and efficiency and innovatively undertakes research on the allocation of carbon emission quotas for railway administrations. This paper constructs an econometric model to analyze the impact of various influencing factors on China's railway operation carbon emission and predicts the total carbon emission of China's railway operation from 2021 to 2030 by scenario analysis method. From the perspective of equity and efficiency, apply the entropy method to give weight to historical responsibility, egalitarianism, and efficiency principle to obtain the initial allocation value of the carbon emission quota of the operator's 18 regional railway bureau groups; the ZSG-DEA model is used to obtain the optimal allocation. The results show that railway passenger turnover, freight turnover, vehicle structure, and per capita GDP have a promoting effect on railway carbon emission, and the proportion of clean energy has an inhibitory effect on carbon emission. There is a gap between the distribution results under the single principle and the comprehensive distribution results; the combination of both can more effectively promote the development of the railway industry. From the perspective of equity and efficiency, the carbon emission quota of 18 railway bureau groups in China is high in the east and low in the west. Among them, the Shanghai railway bureau obtains the most carbon emission quota, while the Qinghai–Tibet railway bureau obtains the least carbon emission quota. The research results provide a reference for the railway bureau to coordinate emission reduction and the construction of the railway transport carbon emission market.

**Keywords:** railway; equity; efficiency; carbon allocation; ZSG-DEA model

## 1. Introduction

In recent years, China's total carbon emissions have surpassed the United States and the European Union to become the world's largest carbon dioxide emitter [1]. China is the world's largest energy consumer, with total energy consumption increasing from 602 million tons of standard coal in 1980 to 4.98 billion tons in 2019, accounting for the largest share of total energy consumption of any country in the world [2]. The transportation industry has developed into the second-largest carbon-emitting industry in China after industry [3]. The Fifth Assessment Report of the IPCC (United Nations Intergovernmental Panel on Climate Change) pointed out that the transportation sector is one of the most energy-consuming industries. The transportation sector has developed into the second largest source of carbon emissions due to the continuous increase in the use of fossil fuels and transportation demands and distances [4].

In the transportation market, there are five general transportation modes, including road, railway, civil aviation, water transportation, and pipeline transportation; road transport produces the highest proportion of carbon dioxide in the entire transport industry [5,6].

In contrast, railways are a relatively environmentally friendly and low-carbon mode of transportation [7,8]. The *14th Five-Year Plan for the Development of Modern Comprehensive Transportation System* issued by the State Council states that "the total mileage of China's railways will reach 175,000 km in 2025 and 200,000 km by 2030." The *Action Plan for Carbon Peaking by 2030* issued by the State Council states that "the comprehensive energy consumption of national railway unit conversion turnover in 2030 will be 10% lower than that in 2020" in China. In 2020, China solemnly promised at the 75th session of the United Nations General Assembly that China's carbon dioxide emissions would peak by 2030 and achieve carbon neutrality by 2060. The State Council has committed to reducing carbon dioxide emissions per unit of GDP by more than 65% by 2030 compared with 2005 in the Action Plan for Carbon Peaking by 2030 in China. Initial carbon emission quota allocation is the premise and basis for carbon trading. Carbon emission trading (CET) has been regarded as an effective tool for achieving such targets. Meanwhile, as a market-oriented emission reduction mechanism, the national emission trading system was established in 2017 on the basis of carbon trading pilots in Shanghai, Beijing, Guangdong, Shenzhen, Tianjin, Hubei, and Chongqing. Railways are seen as vital infrastructure for economic growth, and a perfect railway network enables convenient and rapid exchanges of goods, personnel, information, and elements. In particular, human mobility and cargo transportation are increasingly reliant on high-speed rail in the case of China's vast territory and uneven distribution of energy resources such as coal [9,10]. Sustainability is a strategic choice for the transition to a green economy in China [11], and China is gradually shifting from pursuing rapid economic growth to high-quality economic development quality [12]. This makes it necessary to assign carbon emission quotas for Chinese railway bureau groups, given that China is the No.1 emitter and has by far the largest HSR network in the world.

The economy and development level of Chinese railway bureau groups is not balanced [13]. As a result, there are differences in carbon emissions of Chinese railway bureau groups [14,15]. It has evolved into a major practical challenge to be considered for carbon saving and emission reduction in the railway transport industry, such as how to scientifically measure the carbon emissions of railway transport, how to analyze the main influencing factors of its carbon emission changes, and how to select a reasonable carbon quota allocation mode according to the differences of Chinese railway bureau groups. In this study, our research objective is the allocation of carbon emission quotas in Chinese railway bureau groups.

The contributions of this paper include: (1) Focus on the Chinese railway industry and assigning emission quotas to 18 railway bureaus. Where China's railway sector remains largely state-owned and highly regulated, there are few studies on carbon allocation from the perspective of railway administration. Thus, we provide a reference for the railway industry study of carbon emission quota allocation and facilitating the establishment of a national CET market. (2) Combined with the influencing factors of the railway industry, a carbon allocation model is proposed as a means to trade off efficiency and equity in the allocation of carbon emission quotas, which can make the distribution results fairer and more reliable. (3) This paper first employs the entropy method to do an initial allocation, and then the ZSG-DEA method is used to make the final allocation efficient for all railway bureaus.

The structure of this paper is as follows. Section 2 reviews existing methods and findings by organizing the relevant literature. Section 3 explains the data and methods used in this paper and introduces the framework that we use to assign carbon emission quotas. Section 4 reports and discusses the estimation results. Section 5 provides the findings and conclusions from our research.

## 2. Literature Review

### 2.1. Decomposition of Carbon Emission Reduction Targets

The research of domestic and foreign scholars on the decomposition of carbon emission reduction targets is mainly focused on two aspects. One is the decomposition of carbon

reduction targets at the regional level. Song Jiekun proposed the initial allocation of carbon emission allowances for China's provinces in 2020 and put forward suggestions based on the reduction ratio of carbon emission intensity in each province [16]. Wang Yong made an initial provincial decomposition of China's peak carbon dioxide emissions in 2030 and made it clear that areas with high carbon emissions and high potential were allocated more carbon quotas [17]; Li Jianbao allocated carbon emission quota based on total carbon emissions and intensity constraints in Jiangsu, Zhejiang, and Shanghai in China in 2020 [18]. Qin allocated a carbon emission quota in the eastern coastal areas of China through the multi-criteria decision analysis model, which provides a reference for regionally coordinated emission reduction [19]. On the other hand, the carbon reduction target is broken down at the departmental level. Wang Wanjun made an in-depth analysis of the quotas at all stages of the development of China's industrial system during the 13th Five-year Plan period [20]. Cui adopted the entropy method to study the distribution of carbon emission rights in China's power industry in 2030 [21]. Ji Xiaofeng allocated provincial traffic carbon quotas in China with the 2030 scenario forecast as the total limit and analyzed the provincial traffic carbon reduction pressure [22]. Da Gao proposed that eastern and western cities have a better pollution control effect than the other regions, and large cities have better emission reduction effects than smaller cities [23].

### 2.2. Research on Carbon Emission in Transportation Industry

Up to now, the research on traffic carbon emissions has focused on two levels: one is to measure the carbon emissions of a certain kind of traffic or to compare and analyze the differences in carbon emissions of different transport infrastructures. Chen Jinjie divided the use of high-speed railway life cycle carbon emissions into four stages: building materials production, construction, operation and maintenance, and scrap demolition and disposal, with the largest carbon emissions in the operation and maintenance stages [24]; Wang Chengxin et al. calculated and concluded that among all kinds of transport modes, and the carbon emission of railway transport units is the lowest [25]; Chen Peihong et al. evaluated the net carbon dioxide emissions of Beijing–Shanghai high-speed railway from two aspects of traffic mode substitution and traffic effect through the whole life cycle method [26]; Yu et al. calculated that when the railway operating speed increased by 39.94%, about 104.92 million air passengers turned to rail transport, resulting in a reduction of 13.57 million tons of carbon emissions [27], and rail transport has been regarded as an important means to replace air transport [28,29]. The other is the research on the influencing factors of carbon emissions from transport infrastructure. Zhang Guoxing confirmed that per capita GDP and population size promote carbon emissions, while transportation intensity, energy consumption per unit turnover, and traffic energy intensity inhibit the growth of carbon emissions through the influencing factors of transportation carbon emissions in the Yellow River Basin of China [30]. Gan determined that per capita GDP is the most important stimulus factor for regional carbon emissions, while the tertiary industry and population size inhibited regional carbon emissions by discussing the influence factors of high-speed rail operation on urban carbon dioxide emissions in seven cities in the Hunan section of the Beijing–Guangzhou high-speed railway [31].

### 2.3. Research on Carbon Emission Allocation Quota

The research on the allocation of emission quotas mainly includes equity [32–34], efficiency [35–39], and balance between equity and efficiency [40–45]. Yang Chao studied the distribution of carbon emission rights in China from the perspective of equality, and the historical emission principle can best reflect the principle of equality in regional distribution [33]; Zhou et al. proposed a DEA multi-emission reduction method, constructed a non-radial distance function, proposed a total factor $CO_2$ emission performance index and its dynamic change index to measure $CO_2$ emission performance, and allocated carbon dioxide emission quotas to Chinese cities [39]. How to "take into account" the principle of equity and efficiency is the focus of academia. At present, the most common method is

to use the entropy method to construct a compound index, which eliminates the subjective preference of decision-makers and is easy to operate. Wang Wenju studies the initial carbon quota allocation scheme of provincial regions in China through the combination of equity and efficiency through the entropy method [43]. Lins et al. introduced the zero-sum income model (ZSG-DEA) for the first time. The idea of the model is how to optimize the distribution efficiency of various countries while keeping the total number of medals unchanged [46]. Considering both equity and efficiency, the ZSG-DEA model is widely used in the field of resource allocation [47–51].

There are some deficiencies in the existing relevant research at home and abroad: on the research scale, the existing studies on carbon quota allocation are mainly focused on the decomposition of targets at the national and provincial levels, while there are few studies on the allocation of carbon emission quotas in the transportation industry. From the object of study, most of the research objects of traffic carbon emissions are one kind of transportation mode or comprehensive transportation; no matter which kind of research object, there are few railway carbon emissions allocations from the point of view of the operator's 18 regional railway bureaus. From the research perspective, it is mainly focused on equity, efficiency, and balance between equity and efficiency, while combined with the railway industry's own energy-saving technology level and other factors that affect carbon emissions are considered less. To sum up, there are few studies on carbon allocation in the railway transport industry in the existing literature; moreover, the existing research results of railway carbon emissions mainly focus on the calculation of carbon emissions and the evaluation of low carbon degree at a certain stage of railway operation, which are all based on the posterior analysis of historical data, and lack of scientific and systematic research on the impact factors of carbon emissions and carbon allocation mechanism in the core operation process of modern railways. This indirectly implies that the research on the carbon emission quota of China's railway bureau groups remains in its infancy and has a certain research space and innovation.

Therefore, based on the carbon emission reduction target of China's railway transport industry, combined with the existing research literature on the prediction of China's selected railway operation carbon emissions and economic output level, this paper sets up different scenarios to predict China's railway operation carbon emissions. In addition, under the conditions of determining the total target of carbon emission reduction in the railway transport industry and setting the economic development scenario, the total carbon emission of the railway transport industry as a whole can be calculated and allocated by the railway bureau on this basis. The total carbon emission of the railway bureau is equal to the total carbon emission of the country. This allocation process is similar to the "zero sum game" game theory. Therefore, this paper uses the zero-sum income model is used to allocate carbon to the operator's 18 regional railway bureaus from the perspective of equity and efficiency, with a view to providing policy recommendations for the development of China's railway bureau groups.

## 3. Methodology and Data
### 3.1. Calculation of China's Railway Carbon Emission

The calculation of carbon emission is based on energy consumption. The energy consumption of railway operations mainly includes electricity and diesel. Currently, the conventional treatment method at home and abroad is to convert various energy sources according to energy consumption and standard coal to obtain the total converted energy consumption of standard coal. In this work, a top-down approach is used to measure the carbon emission of railway operation energy consumption:

$$Y = f \times Z \times \frac{44}{12} \tag{1}$$

where, Y represents carbon dioxide emissions; Z represents the consumption of standard coal; f represents the carbon emission coefficient of standard coal, which employs the

reference value proposed by the Japanese Energy Economics Research Institute, that is, 0.68 tons of carbon emission per ton of standard coal.

*3.2. Total Carbon Quota Allocation of China's Railways*

3.2.1. Factors Affecting Carbon Emission of China's Railway Operation

The carbon emission of railway operations is affected not only by the economy and population but also by the railway operation itself. Therefore, passenger turnover, freight turnover, vehicle structure, and the proportion of clean energy and per capita GDP are selected as independent variables. As shown in Table 1, the expression is as follows:

$$\ln Y_{it} = \ln a_0 + a_1 \ln K_{it} + a_2 \ln H_{it} + a_3 \ln VS_{it} \\ + a_4 \ln PC_{it} + a_5 \ln PGDP_{it} + \ln e \tag{2}$$

where $Y_{it}$ represents the carbon emission efficiency of the railway bureau in the t-th year; $K_{it}$, $H_{it}$, $VS_{it}$, $PC_{it}$, respectively, represent the passenger turnover, freight turnover, the ratio of electric locomotives to diesel locomotives, the proportion of electricity in total energy consumption of 18 railway bureau groups; $PGDP_{it}$ represents per capita GDP in the provinces under the jurisdictional territories of 18 railway bureau groups; $a_1$, $a_2$, $a_3$, $a_4$, and $a_5$ are the elastic coefficients of the above variables; $a_0$ is the model coefficient; and $e$ is the random error term.

**Table 1.** Factors influencing carbon emission from railway operations.

| Index Attribute | Index | Definition | Sign |
|---|---|---|---|
| The influence of railway transportation | Passenger turnover | Passenger turnover | K |
| The influence of railway transportation | Freight turnover | Freight turnover | H |
| Energy technology | Vehicle structure | The ratio of electric locomotives to diesel locomotives | VS |
| Energy transformation | The proportion of clean energy | The proportion of electricity in total energy consumption | PC |
| Economic development | Per capita GDP | Ratio of GDP to total population | PGDP |

Based on the econometric model, the carbon emission of railway operation is fitted with various factors by regression equation, and the prediction model of carbon emission of railway operation is constructed, which is combined with scenario analysis.

3.2.2. China's Railway Carbon Emission Scenario Setting

Scenario analysis is widely used to study the future change trend of carbon emission. In this study, three scenarios are established based on the actual situation faced by China's railway operation and the possible future carbon emission: benchmark scenario, energy-saving scenario, and low-carbon scenario.

Benchmark scenario: take economic growth as the main driving factor, follow the law of economic development, and implement it in accordance with conventional development policies.

Energy saving scenario: focus on optimizing the energy structure of the railway industry and technical means to enhance the energy efficiency to achieve the coordinated development of the economy and carbon emission of the railway transport industry.

Low-carbon scenario: the economic development model is more scientific and reasonable, seeks the best way to reduce emissions in the railway transport industry, and adheres to the path of the low-carbon development scenario.

*3.3. Construction of China's Railway Carbon Emission Allowance Allocation Model*

On the basis of ensuring the realization of the carbon emission reduction target of China's railway transport industry, this paper mainly proposes three initial quota allocation schemes, namely, the principle of equity, the principle of efficiency, and the comprehensive principle of taking efficiency and equity into account.

### 3.3.1. Equity and Efficiency-Based Carbon Quota Allocation Model

The principle of historical responsibility is concerned with the equity of the sharing of responsibility for emission reduction by the railway bureau group, while the principle of egalitarianism is concerned with the equality between people, and the two complement each other. In order to achieve more effective incentives to reduce emissions through policy means, the efficiency principle is incorporated into the quota allocation model. This work adopts the principles of historical responsibility, equality, and efficiency to make a preliminary allocation of carbon emission quota for the operation of the railway bureau group. Three indicators of historical cumulative carbon emission, population, and carbon emission efficiency are selected. The principles and implications of each index are shown in Table 2.

**Table 2.** Principles and indicators of carbon emission distribution in railway operations.

| Principle | Concrete Principles | Index | Mean |
|---|---|---|---|
| Equitable principle | Principle of historical responsibility | Historical cumulative carbon emission | Railway bureau group limited with more historical carbon emission allocating more carbon quotas |
| Equitable principle | Principle of egalitarianism | Population | Railway bureau group Co., Ltd. with a larger population in the jurisdiction has more carbon emission rights |
| Efficiency principle | Efficiency principle | Carbon emission efficiency | Railway bureau group limited with low carbon emission efficiency allocating fewer carbon quotas |

The principle of historical responsibility is allocated according to the proportion of the historical cumulative carbon emission of each railway bureau in the country's total cumulative carbon emission. This method ensures the continuity of economic production as far as possible. The calculation formula is as follows.

$$\text{hco}_{2i} = \frac{Y_i}{\sum\limits_{i=1}^{n} Y_i} \times Y_{2030} \quad i = 1, 2, \ldots 18 \tag{3}$$

where $Y_i$ represents the historical carbon emission of the i-th railway bureaus (historical cumulative emissions from 2006 to 2018); $Y_{2030}$ represents the total national emissions in 2030.

The principle of egalitarianism allocates according to the population proportion of each railway bureau group limited company in the base period, which tilts the carbon emission quota to the railway bureaus with a large population in the area under its jurisdiction, which reflects the equality per capita. The calculation formula is as follows:

$$\text{pco}_{2i} = \frac{P_i}{\sum\limits_{i=1}^{n} P_i} \times Y_{2030} \quad i = 1, 2, \ldots 18 \tag{4}$$

where $P_i$ represents the population of the railway bureau in the base period (the resident population in 2018).

According to the efficiency principle, the static $CO_2$ emission efficiency index TCPI is constructed with reference to Zhou, and the non-radial DEA model is used to calculate the efficiency of China's railway bureaus in 2018 (Zhou et al., 2018). Labor, capital, and energy input are expressed by railway employees, railway operating mileage, and energy converted standard coal, respectively, and the expected output is the passenger and freight turnover of each railway bureau, and the non-expected output is carbon emission.

TCPI (efficiency value) is the ratio of the ideal $CO_2$ emission intensity $CI_i^*$ to the actual $CO_2$ emission intensity $CI_t$, so the ideal $CO_2$ carbon emission intensity $CI_i^*$ of the i-th railway bureau is calculated as follows:

$$CI_i^* = CI_i \times TCPI_i \quad i = 1, 2, \ldots 18 \tag{5}$$

In Equation (5), $CI_i^* = CI_i$ for seven railway bureaus located on the production front; while $CI_i^* < CI_i$ for other railway bureaus with low carbon emission efficiency.

If the carbon emission in 2018 is $Y_i$ and the GDP is $GDP_i$, then the annual carbon emission intensity is:

$$CI_i = \frac{Y_i}{GDP_i} \quad i = 1, 2, \ldots 18 \tag{6}$$

Thus, a quota allocation method based on carbon emission efficiency is constructed by using the static total factor $CO_2$ emission efficiency index TCPI:

$$eco_{2i} = \frac{CI_i^* \times GDP_i}{\sum\limits_{k=1}^{n} CI_i^* \times GDP_i} \times Y_{2030} \quad i = 1, 2, \ldots 18 \tag{7}$$

The above three principles reflect a certain sense of equality and efficiency in the process of carbon quota allocation as a scarce resource. The carbon quota allocation of different railway bureaus is equivalent to the allocation of economic development rights, and the optimal carbon emission allocation should consider equality and efficiency.

Therefore, the entropy method is used to integrate the distribution scheme of the historical responsibility principle, egalitarianism principle, and efficiency principle, and the comprehensive quota allocation of railway operation with both equality and efficiency can be obtained according to this method.

$$mco_{2i} = w_1 \times hco_{2i} + w_2 \times pco_{2i} + w_3 \times eco_{2i} \tag{8}$$

In the formula, $w_1$, $w_2$, and $w_3$ are the weights of the historical responsibility principle, egalitarianism, and efficiency principle.

### 3.3.2. ZSG-DEA Model

The ZSG-DEA model has been widely used in the field of resource allocation. Under the condition that the total carbon emission remains unchanged, the increase of carbon emission in one railway bureau is equal to the decrease of carbon emission in other railway bureaus, which reflects the idea of "zero sum game". Combining the "zero sum game" idea with the DEA model, we can build an input-oriented zero-sum income DEA model and reallocate the carbon dioxide emissions of the ineffective railway bureau to achieve the optimal carbon emission efficiency of each railway bureau.

In the ZSG-DEA model, assume that the efficiency value of $DUM_0$ is $\theta_{i0}$, to make $DUM_0$ effective, the investment must be reduced $d_0 = x_{i0}(1 - \theta_{i0})$, and do is allocated to $DUM_i$, according to the proportion of $DUM_i$ in input, $DUM_i$ obtain the allocated input.

$$\frac{x_i}{\sum\limits_{i \neq 0} x_i} x_{i0} \times (1 - \theta_{i0}) \tag{9}$$

where $x_i$ is the input of the i-th $DUM$, and $\theta_{i0}$ is the ZSG-DEA efficiency of the $DUM$.

After all $DUM$ are reduced according to Formula (9), the quota obtained by $DUM_i$, consists of two parts; some are allocated from other $DUM$, the other part is $DUM_i$ to be reduced. The allocation quota is as follows:

$$x_i' = \sum_{i=1}^{N} \lambda_i x_i \times \left[ 1 + \frac{x_{i0} \times (1 - \theta_{i0})}{\sum\limits_{i \neq 0} x_i} \right] \leq \theta_{i0} x_{i0}, \quad i = 1, 2, 3, \ldots, N \tag{10}$$

In order to emphasize the influence of each factor on the carbon emission of railroad operation, the paper takes the carbon emission quota of each railway bureau as the input index of the ZSG-DEA model, introduces passenger turnover, freight turnover, vehicle structure, energy-saving technology level and GDP per capita as output index, and evaluates the initial carbon quota efficiency and adjusts iteratively. The ZSG-DEA model equation is:

$$\begin{cases} \min \theta_{i0} \\ \sum\limits_{i=1}^{N} \lambda_i y_{ij} \geq y_{0j}, \; j = 1, 2, 3, \ldots, M \\ \sum\limits_{i=1}^{N} \lambda_i x_i \times [1 + \frac{x_{i0} \times (1 - \theta_{i0})}{\sum\limits_{i \neq 0} x_i}] \leq \theta_{i0} x_{i0}, \; i = 1, 2, 3, \ldots, N \\ \sum\limits_{i=1}^{N} \lambda_i = 1, \; i = 1, 2, 3, \ldots, N \\ \lambda_i \geq 0, \; i = 1, 2, 3, \ldots, N \end{cases} \quad (11)$$

wherein: $\theta_{i0}$ is the i carbon emission efficiency of the railway bureaus, $\lambda_i$ is the proportion of other the railway bureaus in the effective portfolio of i reconstructing a decision unit relative to the target railway bureau, $y_{ij}$ is the output variable of type j of the i railway bureau, $y_{0j}$ is the value of each output variable of the target railway bureau, $x_{i0}$ and $x_i$ are initial value and adjusted value of the carbon emission allowance of the i railway bureau, and N and M are the number of the Railway Administration Groups and output factors. The best efficiency allocation scheme is obtained through multiple iterations of proportional reduction of input variables.

*3.4. Data Sources*

In the econometric model, the research sample is the 18 railway bureaus in China in 2006–2018, and the original data of total energy consumption (converted to standard coal) were obtained from the *Compilation of Railway Administration Statistics*. Railway passenger turnover, freight turnover, vehicle structure, and the proportion of clean energy were obtained from the *Compilation of Railway Administration Statistics* and *Railway Statistical Bulletin*, and the GDP per capita was obtained from *China Statistical Yearbook*.

The historical cumulative carbon emission in the historical responsibility principle was set as the sum of carbon emissions in 2006–2018, and the original data were obtained from each of the railway bureaus in the *Compilation of Railway Administration Statistics in 2006–2018*.

The population numbers of each railway bureau in the equality principle were obtained from *China Statistical Yearbook in 2018*.

Total energy consumption, railway employees, railway operating mileage, passenger turnover, and freight turnover of each railway bureau in the efficiency principle were obtained from the *Compilation of Railway Administration Statistics in 2018*, and the GDP quantity was obtained from *China Statistical Yearbook in 2018*.

**4. Result and Analysis**

*4.1. Total Carbon Emission of China's Railway in 2030*

4.1.1. Analysis of Influencing Factors on China's Railway Operations

This paper uses Equation (1) to calculate the carbon emission of railway operations in China from 2006 to 2018; the results are shown in Figure 1. According to the slope, it can be divided into three stages: the fluctuation period from 2006 to 2013, the sharp decline period from 2014 to 2015, and the steady rise period from 2016 to 2018. Although China's railway operation carbon dioxide emissions from 2006 to 2018 generally showed a fluctuating upward trend, the average level was maintained at 28,150,000 tons. The trend of carbon emissions and energy consumption is basically the same. This, to a certain extent, indicated that the energy consumption structure of railway operations was improved in China, thus slowing down the growth trend of carbon emissions. In order to explore the

intrinsic influence of carbon emissions of railway transportation, each influencing factor is further analyzed.

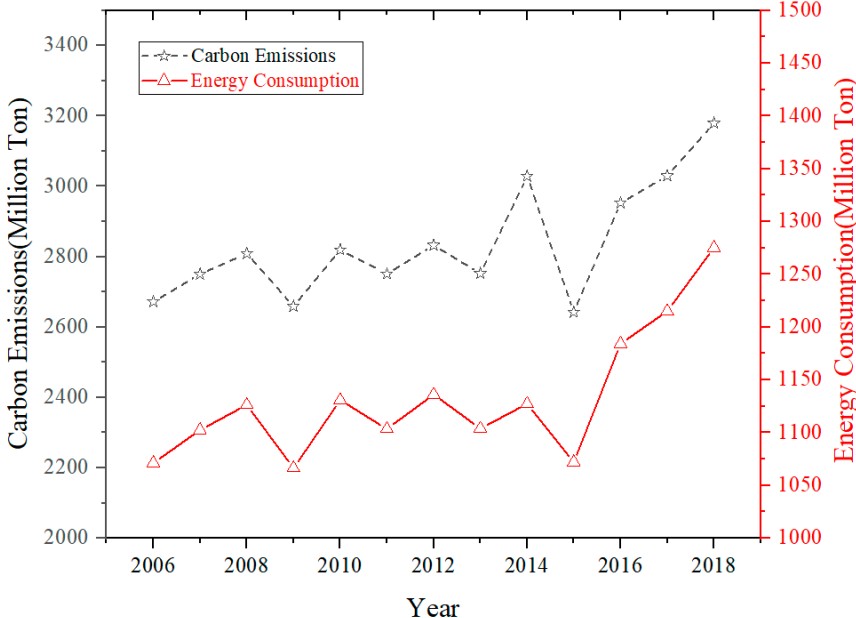

**Figure 1.** Trends of carbon emission and energy consumption in railway operation in China.

To analyze the effects of passenger turnover, freight turnover, vehicle structure, the proportion of clean energy, and GDP per capita of railway bureaus on the total carbon emission of railway operations, a multiple linear regression model was established based on the data of each indicator in 2006–2018 according to Equation (2) to estimate the regression coefficients of each indicator.

VIF value (variance expansion factor) is used to determine whether there is multi-collinearity among the variables (Table 3). The results showed that the VIF values of the five variables of passenger turnover, freight turnover, vehicle structure, the proportion of clean energy, and GDP per capita were all less than 10, and it indicates that there was no multi-collinearity problem among the variables at this time. Table 4 contains mixed-effects regression, fixed-effects regression, and random-effects regression, and it can be concluded by F-test and Hausman test that the fixed-effects model is better than the mixed-effects and random-effects models.

**Table 3.** Multiple collinearity test of carbon emission indexes of railway operation.

| Variable | VIF | 1/VIF |
|----------|------|--------|
| lnk      | 4.29 | 0.2332 |
| lnH      | 4.06 | 0.2463 |
| lnVS     | 1.80 | 0.5569 |
| lnPC     | 1.54 | 0.6494 |
| lnPGDP   | 1.45 | 0.6908 |
| MeanVIF  | 2.63 | -      |

**Table 4.** Estimation results of influencing factors of carbon emission in railway operation.

| Variables | Mixed-Effect | Fixed-Effect | Random-Effect |
|---|---|---|---|
| lnk | 0.337 *** | 0.227 *** | 0.323 *** |
| | (0.0237) | (0.0768) | (0.0425) |
| lnh | 0.476 *** | 0.219 ** | 0.395 *** |
| | (0.0326) | (0.0894) | (0.0561) |
| lnvs | 0.0978 *** | 0.163 *** | 0.149 *** |
| | (0.0206) | (0.0242) | (0.0215) |
| lnpc | −0.482 *** | −0.595 *** | −0.571 *** |
| | (0.0427) | (0.0380) | (0.0382) |
| lnpgdp | 0.146 *** | 0.379 *** | 0.272 *** |
| | (0.0454) | (0.0735) | (0.0541) |
| Constant | 8.254 *** | 10.40 *** | 8.716 *** |
| | (0.205) | (0.729) | (0.374) |
| Observations | 234 | 234 | 234 |
| R-squared | 0.823 | 0.579 | |
| Number of id | 18 | 18 | 18 |

Note: **, *** represent 5% and 1% significance levels. The number in the parenthesis is the estimated standard deviation.

According to the regression results, it can be seen that all five explanatory variables passed the 5% significance level test in the form of:

$$\ln Y_{it} = 0.227 \ln K_{it} + 0.219 \ln H_{it} + 0.163 \ln VS_{it} \\ - 0.595 \ln PC_{it} + 0.379 \ln PGDP_{it} + 10.40 \quad (12)$$

Each 1% increase in railway passenger turnover, freight turnover, vehicle structure, and GDP per capita will increase the carbon emission of railway operations by 0.227%, 0.219%, 0.163%, and 0.379%, respectively. All the above factors will have a contributing effect on railway carbon emission, with GDP per capita having the most significant effect on carbon emission. The share of clean energy plays a suppressive role in railway carbon emission, with the largest negative effect brought by the proportion of clean energy with a coefficient of −0.595%.

There is a long-term equilibrium relationship between the dependent and independent variables, i.e., predicting carbon emission from railway operations requires the future values of the influencing factors.

### 4.1.2. Scenario Analysis on China's Railway Carbon Emission

Scenario analysis is commonly used to study the future trend of carbon emissions. In the paper, we refer to *14th Five-Year Energy Plan* and *14th Five-Year Railway Development Plan* in China and set the detailed parameters of three development scenarios for each factor in 2021–2030 according to Chinese economic and social situation and the railway energy development plan, as show in Table 5.

**Table 5.** Setting the growth rate of each influencing factor.

| Scenario | Year | Passenger Turnover | Freight Turnover | Vehicle Structure | The Proportion of Clean Energy | Per Capita GDP |
|---|---|---|---|---|---|---|
| Benchmark scenario | 2021–2022 | 35.00% | 5.00% | 2.00% | 1.30% | 6.00% |
| | 2023–2025 | 5.00% | 5.00% | 2.00% | 1.30% | 6.00% |
| | 2026–2030 | 4.50% | 3.00% | 1.50% | 1.10% | 5.00% |
| Energy-saving scenario | 2021–2022 | 34.00% | 4.50% | 1.50% | 1.10% | 5.50% |
| | 2023–2025 | 4.50% | 4.50% | 1.50% | 1.10% | 5.50% |
| | 2026–2030 | 4.00% | 2.50% | 1.00% | 0.90% | 4.50% |
| Low-carbon scenario | 2021–2022 | 33.00% | 4.00% | 1.00% | 0.90% | 5.00% |
| | 2023–2025 | 4.00% | 4.00% | 1.00% | 0.90% | 5.00% |
| | 2026–2030 | 3.50% | 2.00% | 0.50% | 0.70% | 4.00% |

Combined with the multi-scenario analysis, the total transportation carbon emission projections in 2021–2030 under the multi-scenario are shown in Figure 2. In the baseline scenario, carbon emissions from the railway operations will reach 129,735,724 tons in 2030; In the energy-saving scenario, the growth slowdown of carbon emissions from the railway operations will reach lower 126,055,476 tons in 2030; In the low carbon scenario, carbon emissions from railway operations will reach 119,816,606 tons in 2030, a reduction of 9,919,118 tons than the baseline scenario. In order to promote the green development of low-carbon railways and achieve the carbon peak vision as soon as possible, the carbon emission under the low-carbon scenario in 2030 is used as the total amount of quotas to control the carbon emission from the railway operations through coordinated development among the railway bureaus.

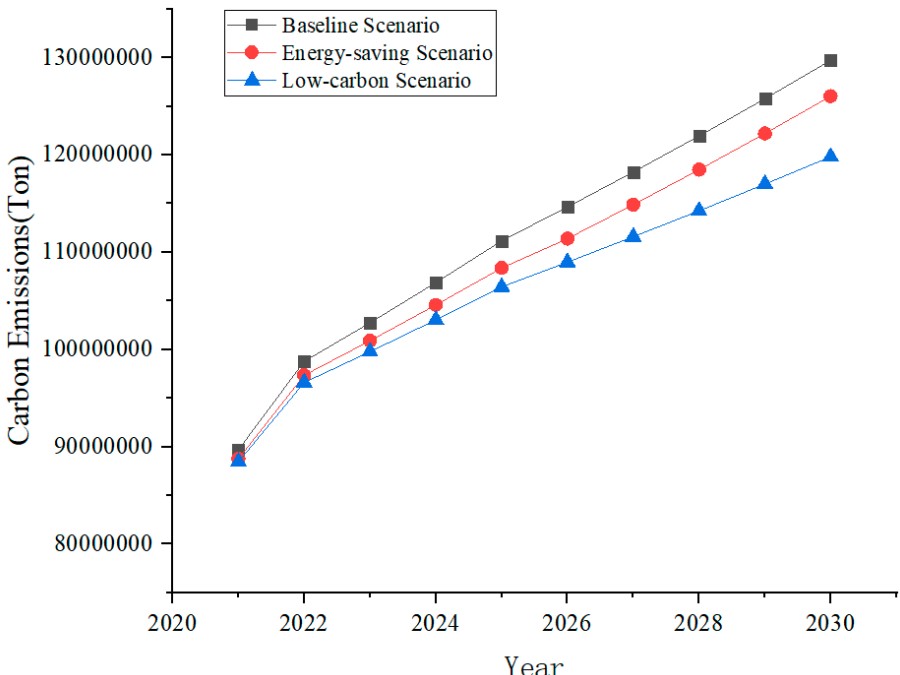

**Figure 2.** Prediction of total carbon emission in railway operation under different scenarios.

*4.2. Allocation Model on Carbon Quota*

4.2.1. Results of Quota Allocation Based on Single Model

The allocation results based on the three single principles of historical responsibility, equality, and efficiency were calculated according to Equations (3), (4), and (7) in Figure 3.

Based on the principle of historical responsibility, Shenyang, Shanghai, Beijing, and Harbin railway bureaus share the most carbon quotas in turn, while Qinghai–Tibet and Kunming railway bureaus share only 5,923,771 and 5,986,378 tons of quotas. In 2018, the freight turnover of Shenyang railway bureau was 237.1 billion ton-kilometers, and the passenger turnover of Shanghai railway bureau were 241 billion person-kilometers, but the passenger turnover and the freight turnover of Qinghai–Tibet railway bureau were 10.9 billion person-kilometers and 30.8 billion ton-kilometers, respectively, which shows that the energy consumption due to the increase of workload is the fundamental reason for the change of carbon emission quota of the railway bureau groups.

Based on the principle of equality, the Qinghai–Tibet railway bureau, which is located in the Great Northwest Economic Zone, had a resident population of 9.47 million in 2018, and it has the least amount of carbon quotas according to population indicators. The provinces with higher quota allocations are mostly concentrated in the railway bureau groups located in the southeastern region, with the Shanghai railway bureau having the largest population under its jurisdiction with a resident population of 225.36 million at the end of 2018, 23 times that of Qinghai–Tibet railway bureau.

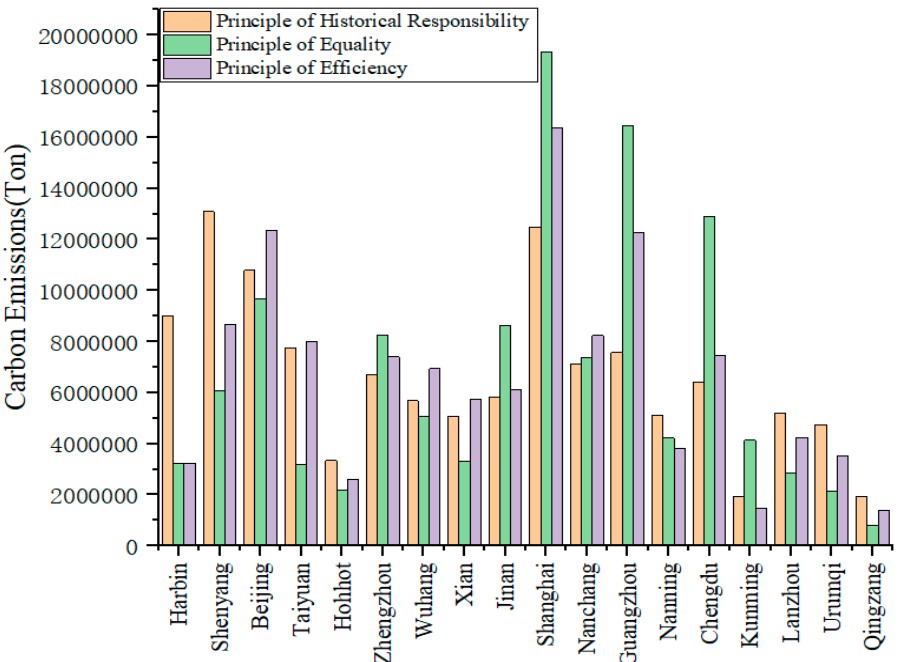

**Figure 3.** Initial carbon allocation results of railway bureau based on single principle.

Based on the efficiency principle, which showed the $CO_2$ emission efficiency index of China State Railway Group in 2018 as Figure 4. There are the seven railway bureaus, such as Shanghai, Beijing, Guangzhou, and Taiyuan Railway railway bureau, which are located at the front surface of production, and they are all located in the core area of the railway network, linking several important railway corridors with frequent economic exchanges. While Harbin, Qinghai–Tibet, and Kunming railway bureaus are in the last three places in the $CO_2$ emission efficiency index. Due to poor weather conditions and high input and operation costs, the demand output is low, and the spatial difference in the carbon emission efficiency of the railway bureau is obvious.

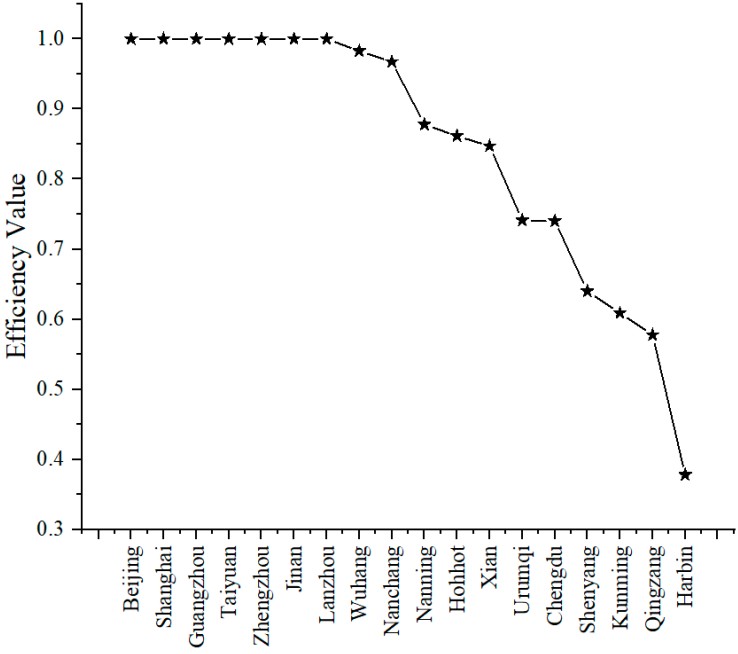

**Figure 4.** Efficiency values of railway bureaus in 2018.

### 4.2.2. Initial Allocation Results of Carbon Quota Based on Equity and Efficiency

The weight of the three allocation methods is calculated based on the principle of historical responsibility, the principle of egalitarianism, and the principle of efficiency by Equation (8), and the calculation results are shown in Table 6.

**Table 6.** Weights of three single principle allocation methods.

| Weight | Principle | | |
|---|---|---|---|
| | **Historical Responsibility** | **Egalitarianism** | **Efficiency** |
| | 0.2722 | 0.3888 | 0.3390 |

The allocation results of the three independent principles are very different. Taking the Harbinrailway bureau as an example, the largest quota is allocated under the principle of historical responsibility in 2030, and it reaches 9,003,319 tons, while the allocation results under the two principles of egalitarianism and efficiency are 3,237,081 tons and 3,229,246 tons, respectively. The allocation of the comprehensive principle that considers efficiency and equity reduces the difference between the three at 4,804,078 tons, as shown in Table 7.

**Table 7.** Results of carbon quota distribution of China railway bureaus in 2030.

| Railway Bureau | Carbon Quota Allocation | | | |
|---|---|---|---|---|
| | **Historical Responsibility** | **Egalitarianism** | **Efficiency** | **Comprehensiveness** |
| Harbin | 9,003,319 | 3,237,081 | 3,229,246 | 4,804,078 |
| Shenyang | 13,083,569 | 6,059,767 | 8,659,444 | 8,853,033 |
| Beijing | 10,783,181 | 9,669,203 | 12,345,228 | 10,879,612 |
| Taiyuan | 7,767,680 | 3,189,893 | 8,013,547 | 6,071,243 |
| Hohhot | 3,342,397 | 2,174,069 | 2,618,146 | 2,642,646 |
| Zhengzhou | 6,689,366 | 8,240,700 | 7,395,018 | 7,531,720 |
| Wuhang | 5,685,413 | 5,076,546 | 6,938,908 | 5,873,626 |
| Xian | 5,086,257 | 3,315,155 | 5,746,051 | 4,621,344 |
| Jinan | 5,832,440 | 8,619,918 | 6,115,880 | 7,012,262 |
| Shanghai | 12,486,896 | 19,334,973 | 16,364,506 | 16,463,845 |
| Nanchang | 7,113,175 | 7,369,013 | 8,234,133 | 7,592,645 |
| Guangzhou | 7,573,345 | 16,454,804 | 12,258,017 | 12,614,440 |
| Nanning | 5,121,700 | 4,226,308 | 3,810,665 | 4,329,144 |
| Chengdu | 6,422,297 | 12,906,283 | 7,451,280 | 9,292,012 |
| Kunming | 1,947,998 | 4,143,944 | 1,463,694 | 2,637,576 |
| Lanzhou | 5,204,537 | 2,852,715 | 4,247,834 | 3,965,858 |
| Urumqi | 4,745,409 | 2,133,745 | 3,531,524 | 3,318,522 |
| Qingzang | 1,927,625 | 812,488 | 1,393,485 | 1,313,001 |
| Sum | 119,816,606 | 119,816,606 | 119,816,606 | 119,816,606 |

### 4.2.3. Optimal Adjustment of Carbon Quota Redistribution

According to the share of carbon quota of the comprehensive principle of each railway bureau, the total projected low carbon scenario is initially allocated proportionally in 2030. Taking the passenger turnover, freight turnover, vehicle structure, the proportion of clean energy, and GDP per capita of each railway bureau as output in 2030 and the initial carbon quota value as input, the initial carbon quota efficiency of each railway bureau is calculated.

The results show that in the initial allocation case, among the 18 railway bureau groups, 13 Groups have an initial allocation efficiency value of one. Among them, the initial efficiency value of five of the railway bureaus of China State Railway Group, such as Harbin, Shenyang, Nanchang, Nanning, and Urumqi, do not reach one. Harbin railway bureau is the lowest, followed by Nanjing railway bureau with efficiency values of 0.6358 and 0.8118, respectively.

Since the initial allocation results, the carbon emission efficiency did not reach one in some of the railway bureau groups; there is still room for improving the allowance efficiency. The ZSG-DEA model is constructed to make the carbon emission efficiency reach one by iterative calculation in each railway bureau, and the iterative adjustment process is shown in Table 8. The results show that after three iterative adjustments, the carbon emission efficiency of China State Railway Group is optimal. The 13 railway bureau groups with higher initial efficiency values and their carbon emission quota have increased; the railway bureaus with lower initial efficiency values have increased their efficiency values by splitting the input indexes to other the railway bureau groups, and their carbon emission quota has decreased, among which Harbin railway bureau has the most obvious decrease, as shown in Figure 5.

After optimization and adjustment, the carbon quota of railway bureaus show a spatial trend decreasing from the more economically developed eastern regions to the less developed western regions and from the places with abundant coal resource production to the places with energy demand. There is high population density, a developed economy, and a dense road network in the Shanghai, Beijing, and Guangzhou railway bureaus of China with higher efficiency of the carbon emissions of the of railway operations, so they obtain larger carbon emission quotas and form obvious high-value areas. Jilin and Liaoning provinces under the jurisdiction of the Shenyang railway bureau are traditional energy industry bases and resource-based regions, rich in coal and other resources, which need to be transported to other provinces and cities by railway with convenient transportation locations, thus gaining more carbon quotas. In 2030, the Shanghai railway bureau received the most carbon emission quota at 17,023,921 tons, accounting for 14.21% of the total quota. Qinghai–Tibet, Hohhot, and Kunming railway bureaus are located in western China and have a large area under their jurisdiction, but they are located in the plateau or desert area with railway transportation facilities to be improved, and the economy is relatively backward, so they obtain fewer carbon quotas. The Qinghai–Tibet railway bureau received the lowest railway carbon emission quota at 1,357,668 tons, accounting for 7.98% of the Shanghai railway bureau.

**Table 8.** Carbon emission quota of railway operation in 2030 based on ZSG-DEA Model.

| Railway Bureau | 2030 Carbon Emissions (Before Optimization)/Ton | Incipient | The Frist | The Second | The Third | 2030 Carbon Quota (AfterOptimization)/Ton | Adjustment Amount (Tons) |
|---|---|---|---|---|---|---|---|
| | | Efficiency Value | Iteration | Iteration | Iteration | | |
| Harbin | 4,804,078 | 0.6358 | 0.9999 | 0.9999 | 1 | 3,110,790 | −1,693,288 |
| Shenyang | 8,853,033 | 0.9260 | 0.9999 | 0.9999 | 1 | 8,443,492 | −409,541 |
| Beijing | 10,879,612 | 1 | 1 | 1 | 1 | 11,249,720 | 370,108 |
| Taiyuan | 6,071,243 | 1 | 1 | 1 | 1 | 6,277,740 | 206,497 |
| Hohhot | 2,642,646 | 1 | 1 | 1 | 1 | 2,732,545 | 89,899 |
| Zhengzhou | 7,531,720 | 1 | 1 | 1 | 1 | 7,787,939 | 256,219 |
| Wuhang | 5,873,626 | 1 | 1 | 1 | 1 | 6,073,438 | 199,812 |
| Xian | 4,621,344 | 1 | 1 | 1 | 1 | 4,778,556 | 157,212 |
| Jinan | 7,012,262 | 1 | 1 | 1 | 1 | 7,250,809 | 238,547 |
| Shanghai | 16,463,845 | 1 | 1 | 1 | 1 | 17,023,921 | 560,076 |
| Nanchang | 7,592,645 | 0.9586 | 0.9999 | 0.9999 | 0.9999 | 7,513,409 | −79,236 |
| Guangzhou | 12,614,440 | 1 | 1 | 1 | 1 | 13,043,564 | 429,124 |
| Nanning | 4,329,144 | 0.8118 | 0.9934 | 0.9999 | 1 | 3,605,218 | −723,926 |
| Chengdu | 9,292,012 | 1 | 1 | 1 | 1 | 9,608,085 | 316,073 |
| Kunming | 2,637,576 | 1 | 1 | 1 | 1 | 2,727,302 | 89,726 |
| Lanzhou | 3,965,858 | 1 | 1 | 1 | 1 | 4,100,756 | 134,898 |
| Urumqi | 3,318,522 | 0.9150 | 0.9971 | 1 | 1 | 3,131,657 | −186,865 |
| Qingzang | 1,313,001 | 1 | 1 | 1 | 1 | 1,357,668 | 44,667 |
| Sum | 119,816,606 | - | - | - | - | 119,816,607 | - |

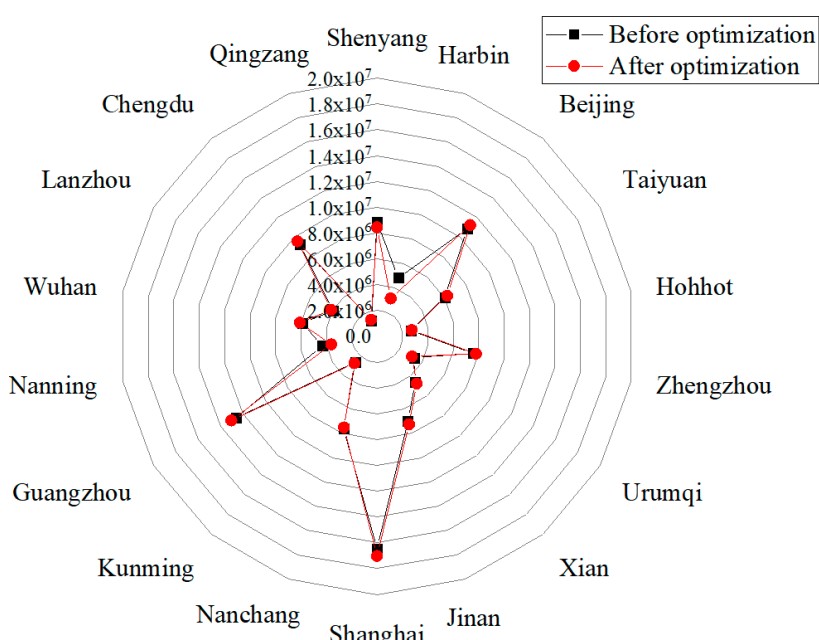

**Figure 5.** Adjustment results of railway bureaus' allocation of carbon emission quota.

## 5. Conclusions and Implication

Based on a more comprehensive accounting of carbon emission of Chinese railroad operations from 2006–2018, the paper first proposes a more scientific and reasonable target for carbon quota for the Chinese railway industry in 2030 using econometric models and scenario forecasts and then analyzes the carbon emission allocation of railway bureau groups value optimization by the perspective of equity and efficiency. A ZSG-DEA model was used to optimize the carbon quota efficiency value in order to provide a reference for the future carbon quota allocation and low-carbon development of the Chinese railway industry. We found that:

Firstly, the proportion of clean energy has the most significant effect on reducing carbon emissions from railway operations, and the elasticity is −0.595%; railway passenger turnover, freight turnover, vehicle structure, and GDP per capita will contribute to the carbon emission of railway operations. The main driving factor for the growth of carbon emissions in railway operations is the continuous growth of per capita GDP, and the main driving factor for effectively inhibiting the growth of carbon emissions is the increasing proportion of clean energy.

Secondly, different allocation mechanisms will produce different policy effects. The historical cumulative carbon emissions of the Shenyang bureau are the highest, and the allocation quota is 13,083,569 tons under the principle of historical responsibility, while the allocation results under the principles of egalitarianism and efficiency are 6,059,767 tons and 8,659,444 tons, respectively. The comprehensive principle of efficiency and equity reduces the difference between the three principles. Since energy structure, development pattern, the population within the jurisdiction, and carbon emission efficiency of each of the railway bureau groups are different, if such regional differences of the railway bureau groups are ignored in the carbon quota allocation mechanism, it will further aggravate the development imbalance within the railway bureau groups.

Thirdly, the comprehensive principle of carbon emission quota results from the perspective of equity and efficiency, which showed that the quota of the railway bureaus companies in western China (such as Qinghai–Tibet and Kunming) is smaller than that of the railway bureaus in east and central China (such as Shanghai, Beijing, and Guangzhou). This is due to the relatively developed economy, complete railway facilities, and high carbon emission efficiency of the railway bureau groups located in east and central China.

The above research findings put forward policy suggestions to improve the carbon market allocation mechanism of the railway transportation industry and promote the low-carbon economic transition. In response to the above research findings, the paper puts forward the following recommendations:

Firstly, reducing carbon emissions through better management of the railway bureau. Improving the share of clean energy, the carbon emission of clean energy electricity is much lower than that of other energy sources, and new energy-saving technologies should be developed and promoted to gradually improve the efficiency of electric energy utilization. Meanwhile, the transportation structure should be optimized, and electric locomotives should be accelerated to reduce locomotive energy consumption. Chinese railway bureau groups carry the responsibility of transport transfer, and the passenger turnover and freight turnover factors have a strong role in promoting the current carbon emission of railroad transport. Reducing the level of energy consumption per unit of turnover is a breakthrough to solve the current transport energy consumption of the national railway transport industry, and we should increase the research and development of energy-saving equipment for railroad transport and reduce the level of energy consumption per unit of turnover.

Secondly, the optimal carbon emission allocation of the railway bureaus groups should be formulated with a balance of economic efficiency and social equity. The scheme provided in this paper can be used as a way to design carbon quotas. Starting from the emission reduction target of the railway transport industry, the emission responsibility, population size, and economic efficiency of railway bureaus are included in the carbon quota allocation mechanism. Resources are reasonably allocated through the price adjustment mechanism to stimulate railway bureaus to participate in trading vitality and promote the economic transformation and upgrading of China's transport industry.

Thirdly, each railway bureau group should take relevant emission reduction measures in conjunction with the carbon quota. The overall spatial distribution of carbon emission quota in the Chinese railroad transportation industry is high in the east and low in the west; there are relatively large differences in the distribution results among the railway bureau groups, so regional differences must be considered when formulating policies. Focus on controlling the hot spots of carbon emission, and formulate emission reduction measures for the central and western railway bureaus according to the development needs. According to the characteristics of carbon emission and its influencing factors, each company should give full play to the synergistic effect among the regions of railway bureau groups focusing on the key work of energy conservation and emission reduction, improving the efficiency of railway carbon emission and further promoting the process of energy conservation and emission reduction in the railway transportation industry.

Although this paper comprehensively analyzes the carbon amount between railway bureau, there are still some limitations, which could also be possible future research directions:

For example, our regression on carbon emissions does not account for any spatial interdependence of the railway bureaus. It seems natural for rail operations to cross the territories of several railway bureaus so that the carbon emission of neighboring bureaus may be correlated. The spatial dependence in railway bureaus can be explicitly accounted for in a future study. Additionally, due to the availability of railway data, the time window of this paper is relatively old, carbon allocation cost of the railway bureau is not considered. Therefore, there are still many problems related to the marginal carbon emission reduction cost to be further analyzed.

**Author Contributions:** Funding acquisition, Z.L.; Methodology, Y.G.; Project administration, Q.T.; Software, Z.L.; Supervision, Q.T.; Visualization, Y.Z.; Writing—original draft, Y.G. All authors have read and agreed to the published version of the manuscript.

**Funding:** This research was funded by The National Natural Science Foundation of China (52102472), the Hebei provincial human resources and social security department project, and the Hebei provincial high-level talent funding project (A202001106), Research topic of social development in Hebei Province (20210201161), Hebei Higher Education Teaching Reform Research Project (2020GJJG390).

**Institutional Review Board Statement:** Not applicable.

**Informed Consent Statement:** Not applicable.

**Data Availability Statement:** Not applicable.

**Acknowledgments:** We would also like to express our thanks to Beijing Jiaotong University and Hengshui University for their assistance in this study.

**Conflicts of Interest:** The authors declare no conflict of interest.

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
