# Peer review of "Research on Carbon Emission Quota of Railway in China from the Perspective of Equity and Efficiency"

_sustainability, doi:10.3390/su142113789_

Round 1

Reviewer 1 Report

After reading this manuscript-193923, I think the authors well-prepared for their content but do not format correctly. I am OK with methodology, data, data analysis and their conclusions.

The manuscript analyzes the impact of various influencing factors on China's railway operation carbon emission and predicts the total carbon emission of China's railway operation from 2021 to 2030 by scenario analysis method. The overall content is fine.

1. The author should follow "guideline for authors" to prepare this manuscript, including citation format, reference format, section format, etc. Current format requires adjustment before publishing.

2. The authors should provide DOIs for all reference.

3. Lines 121, 122, 376 and 381, "CO2" is too big.

4. All equations, from Equation (1) to Equation (9), need to follow mathematical format. 

5. Last term of Equation (2) is ln e, which should equal to 1 if e is denoted as exponential.

6. Line 172, 173 and 175, the variables are too big.

7. It is recommended that the authors should improve the quality of this manuscript for your revision.

8. Writing needs refinement. The whole manuscript requires serious revisions.   9. Section 1 and Section 2 should be merged.   10. The authors should explain "carbon quota allocation model" and "ZSG-DEA model" in more details.   11. How confidence for your prediction accuracy from 2022 to 2030 in China?   12. The authors should add research scope, research assumptions and research limitations in Section 1.

Author Response

Response to reviewer #1:

We would like to thank you for your useful work to our manuscript “Research on Carbon Emission Quota of Railway in China from the Perspective of Equity and Efficiency”. All the comments and suggestions in the 1st round review have been considered by the authors to modify and improve the original manuscript. According to the received review results, we document how each comment was considered and addressed. We refer to the location of the changes in the paper, and the changed sentences and paragraphs have been highlighted in red text. The page numbers in the manuscript where we have addressed each comment are also indicated after response. We hope the editors and reviewers will find that this manuscript has been properly revised and improved:

1. The author should follow "guideline for authors" to prepare this manuscript, including citation format, reference format, section format, etc. Current format requires adjustment before publishing.

Response: Thanks for the valuable suggestion. We have followed "guideline for authors" to revise this manuscript, including citation format, reference format, section format, etc. Meanwhile, we have corrected the formatting errors of all references in page 17 to 20. Incorrect use of articles, typo errors, phrases, grammatical and punctuation-related mistakes have been corrected. (Page 17-20)

2.The authors should provide DOIs for all reference.

Response: Thank you for your suggestions. We have provided DOIs for all reference and highlighted in red type in the revised manuscript. (Page 17-20)

3. Lines 121, 122, 376 and 381, "CO2" is too big.

Response: Thank you for your suggestions. We have and highlighted in red type in the revised manuscript. (Section 3.2, Line 266-281, Page7)

 4. All equations, from Equation (1) to Equation (9), need to follow mathematical format.

Response: Thank you for correcting the errors in our text. We have followed mathematical format to modify all equations, from Equation (1) to Equation (12). (Section 3.2, Page 6-8)

5. Last term of Equation (2) is ln e, which should equal to 1 if e is denoted as exponential.

Response: We are grateful for the suggestion. This paper ln e is denoted as the purely random error term, which take logarithm of error term to reduce the influence of heteroscedasticity between independent variables. (Section 3.1, Line 215, Page 5) 

6. Line 172, 173 and 175, the variables are too big.

Response: Thank you for correcting the errors in our text. We have carefully followed correct format to modify all variables in the revised version. The number of lines have changed due to content adjustment. The number of new lines where the variable is located has been changed in the revision. (Section 3.1, Line 209, 210 and 214, Page 5) 

7. It is recommended that the authors should improve the quality of this manuscript for your revision.

Response: We are grateful for the suggestion. We have carefully proofread in the revised version and improved the quality of this manuscript for your revision. 

8. Writing needs refinement. The whole manuscript requires serious revisions. 

Response: The suggestion is of great significance to us. We have seriously refined the paper and revised the whole manuscript. 

9. Section 1 and Section 2 should be merged. 

Response: Thank you for your suggestions on the introduction and literature review, it has been very enlightening. We have refined the Section 1 and Section 2 after highlighted the background, and we have proposed the methods and summary of innovations of this paper. Meanwhile, we have added the fundamental of statement of the literature and introduced the deficiencies of the existing literature to echo the highlights of the author's literature. (Page 1-4) 

10. The authors should explain "carbon quota allocation model" and "ZSG-DEA model" in more details. 

Response: We are grateful for the suggestion. We have explained "carbon quota allocation model" and "ZSG-DEA model" in more details. (Section3.3 Page 6-8)

11.How confidence for your prediction accuracy from 2022 to 2030 in China?

Response: Thank you for your question. We believe that prediction accuracy from 2022 to 2030 in China for the following reasons. First, our R square is greater than 0.5. When R value is ± 0.7 or more, the two variables are highly correlated, that is, strong correlation; When R value is between ± 0.5 and ± 0.7, the two variables are moderately correlated; When R value is between ± 0.3 and ± 0.5, the two variables are weakly correlated; When R value is lower than ± 0.3, it indicates that there is almost no correlation between the two variables. Second, i th the increase of variables, the R value will not decrease but will increase. Woodridge made it clear that the absolute coefficient is only one of the indicators to judge the quality of the model, not all. Therefore, our variables are relevant and the prediction is reliable. (Section 4.2.1 Table 4)

Table 4. Estimation results of influencing factors of carbon emission in railway operation.

Variables

Mixed-effect

Fixed-effect

Random-effect

lnk

0.337***

0.227***

0.323***

(0.0237)

(0.0768)

(0.0425)

lnh

0.476***

0.219**

0.395***

(0.0326)

(0.0894)

(0.0561)

lnvs

0.0978***

0.163***

0.149***

(0.0206)

(0.0242)

(0.0215)

lnpc

-0.482***

-0.595***

-0.571***

(0.0427)

(0.0380)

(0.0382)

lnpgdp

0.146***

0.379***

0.272***

(0.0454)

(0.0735)

(0.0541)

Constant

8.254***

10.40***

8.716***

(0.205)

(0.729)

(0.374)

Observations

234

234

234

R-squared

0.823

0.579

Number of id

18

18

18

Note: **, *** represent 5% and 1% signifcance levels. The number in the parenthesis is the estimated standard deviation.

 12. The authors should add research scope, research assumptions and research limitations in Section 1.

Response: Thank you for your insightful comment. We have rewritten the contributions and the structure of this paper in section 1, where three points are highlighted and summarized. The practical implications of the research scope, research assumptions and research limitations are also discussed in the introduction. (Section 1, Line 80 -94, Page 2)Other amendments that we think need to be improved are highlighted in red, such as abstract, supplement to the introduction and basic assumption, etc.

Reviewer 2 Report

Reviewer comments:

This paper constructs an econometric model to analyze the impact of various influencing factors on China's railway operation carbon emission and predicts the total carbon emission of China's railway operation from 2021 to 2030 by scenario analysis method. The topic is interesting, but some problems need to be improved. The research object and perspective selected in this paper are good. If the author can modify it according to the opinions, I think the article can be published

1. The abstract is not concise enough but should highlight the methods, results, and innovations. I hope the author will further refine the core and simplify the abstract

2. The structure of the introduction is not clear and fluent enough. After introducing the background, the author should focus on the theme of this paper and propose the methods and innovations of this paper. However, I did not see any summary of the innovations of this paper

3. The literature review part is only a simple arrangement statement of the literature without analysis and induction. It is necessary to point out the deficiencies of the existing literature to echo the highlights of the author's literature.

There are too many literature sections, I suggest the author to summarize in the form of a table, and the following latest relevant articles can also be cited.

Does digitization improve green total factor energy efficiency? Evidence from Chinese 213 cities[J]. Energy, 2022: 123395.

Does FDI improve green total factor energy efficiency under heterogeneous environmental regulation? Evidence from China. Environmental Science and Pollution Research, 1-14.

Dynamic environmental regulation threshold effect of technical progress on green total factor energy efficiency: evidence from China[J]. Environmental Science and Pollution Research, 2022, 29(6): 8804-8815.

Can pollution charges reform promote industrial SO2 emissions reduction?—Evidence from 189 China’s cities[J]. Energy & Environment, 2021, 32(1): 96-112.

4. The format of the formula is incorrect, and garbled characters appear in some formulas

5. As far as I know, the traditional ZSG⁃DEA model automatically assumes that all DMUs have the same technical level. Is there a large bias in the estimation of carbon quota due to the technology heterogeneity ignored by the model? How to prove the robustness of the conclusion?

6.In the Conclusion chapter, interpreting the findings of this study in reflection of critical overview is handled, but very weak. The interpretation is important to highlight the contribution of this study. Please rewrite this part with careful manner. 

7.The implication section must be edited to communicate the novelty of the study and how findings can be applied and used in practical terms.

8.The author needs the full text to be further polished and highlighted.

Author Response

Response to reviewer #2:

We would like to thank you for your useful work to our manuscript “Research on Carbon Emission Quota of Railway in China from the Perspective of Equity and Efficiency”. All the comments and suggestions in the 1st round review have been considered by the authors to modify and improve the original manuscript. According to the received review results, we document how each comment was considered and addressed. We refer to the location of the changes in the paper, and the changed sentences and paragraphs have been highlighted in red text. The page numbers in the manuscript where we have addressed each comment are also indicated after response. We hope the editors and reviewers will find that this manuscript has been properly revised and improved:

1.The abstract is not concise enough but should highlight the methods, results, and innovations. I hope the author will further refine the core and simplify the abstract.

Response: Thank you for your insightful comment. We have rewritten the Abstract to emphasize the methods, results, contribution and novelty of this study. (Page1)

2.The structure of the introduction is not clear and fluent enough. After introducing the background, the author should focus on the theme of this paper and propose the methods and innovations of this paper. However, I did not see any summary of the innovations of this paper.

Response: Thanks for the advice on the structure. We have rewritten the contributions and the structure of this paper in section 1, where three points are highlighted and summarized. In the contributions of this paper, We have proposed the methods and summary of innovations of this paper. (Section 1, Line 80 -94, Page 2).

3.The literature review part is only a simple arrangement statement of the literature without analysis and induction. It is necessary to point out the deficiencies of the existing literature to echo the highlights of the author's literature. There are too many literature sections, I suggest the author to summarize in the form of a table, and the following latest relevant articles can also be cited.

Does digitization improve green total factor energy efficiency? Evidence from Chinese 213 cities[J]. Energy, 2022: 123395.

Does FDI improve green total factor energy efficiency under heterogeneous environmental regulation? Evidence from China. Environmental Science and Pollution Research, 1-14.

Dynamic environmental regulation threshold effect of technical progress on green total factor energy efficiency: evidence from China[J]. Environmental Science and Pollution Research, 2022, 29(6): 8804-8815.

Can pollution charges reform promote industrial SO2 emissions reduction?—Evidence from 189 China’s cities[J]. Energy & Environment, 2021, 32(1): 96-112.

Response: Thank you for your suggestions on the literature review, it has been very enlightening. We have rewritten the literature review in the Introduction section. Meanwhile we have added the fundamental of statement of the literature instead of the form of a table and introduced the deficiencies of the existing literature to echo the highlights of the author's literature. The suggested IoT-related papers has been cited as references 2 ,11,12 and 24 in the revised version, which are also supplemented in the Introduction section. (Page 1-3)

4.The format of the formula is incorrect, and garbled characters appear in some formulas.

Response: Thank you for your comments. We have edited the format of the formula in the revised manuscript. Meanwhile ,we have corrected the formatting errors of all references in page 17 to 20. Incorrect use of articles, typo errors, phrases, grammatical and punctuation-related mistakes have been corrected. (Page 17-20)

5. As far as I know, the traditional ZSG⁃DEA model automatically assumes that all DMUs have the same technical level. Is there a large bias in the estimation of carbon quota due to the technology heterogeneity ignored by the model? How to prove the robustness of the conclusion?

Response: Thank you for your question. China’s railway sector remains largely state owned and highly regulated. Pricing and investment decisions had, until recently, been tightly controlled by the central government in (presumed) efforts to maximize social welfare. The State Railway Administration has committed to issue Railway Industry Standards, which the railway industry standards have been approved and issued by China Railway Administration and are hereby promulgated. Due to the particularity of the railway transport industry, the ZSG⁃DEA model automatically assumes that all DMUs have the same technical level in railway industry.

6.In the Conclusion chapter, interpreting the findings of this study in reflection of critical overview is handled, but very weak. The interpretation is important to highlight the contribution of this study. Please rewrite this part with careful manner. 

Response: Thank you for your suggestions. We have considered the impact of carbon emission quota of railway and we have rewritten the Conclusion chapter to emphasize the contribution of this study. (Section 5, Line 561-580).

7.The implication section must be edited to communicate the novelty of the study and how findings can be applied and used in practical terms.

Response: Thank you for your insightful comment. In the revised version, we have polished and highlighted the implication section which can communicate the novelty of the study and how findings can be applied and used in practical terms (Section 5, Line 581-591).

 8.The author needs the full text to be further polished and highlighted.

Response: We are grateful for the suggestion. We have improved the English writing, and conducted the manuscript carefully in the revised version.

Other amendments that we think need to be improved are highlighted in red, such as citation format, reference format, section format and DOIs for all reference, etc.

Round 2

Reviewer 2 Report

The author has revised it according to my requirements, and the quality of the article has been further improved.

Author Response

We would like to thank you for your useful work to our manuscript “Research on Carbon Emission Quota of Railway in China from the Perspective of Equity and Efficiency”. All the comments and suggestions in the 2ed round review have been considered by the authors to modify and improve the original manuscript. We have studied reviewers' comments point by point, revised the English language and style based on reviewers' comment in the revised manuscript. The amendments based on reviewers' comments are highlighted in red type in the revised manuscript. We refer to the location of the changes in the paper, and the changed sentences and paragraphs have been highlighted in red text. We hope the editors and reviewers will find that this manuscript has been properly revised and improved.

Other amendments that we think need to be adjusted are highlighted in red, such as we the Fundamental Research Funds number., etc.
